# Adsorption of Phenol and Chlorophenols by HDTMA Modified Halloysite Nanotubes

**DOI:** 10.3390/ma13153309

**Published:** 2020-07-24

**Authors:** Piotr Słomkiewicz, Beata Szczepanik, Marianna Czaplicka

**Affiliations:** 1Institute of Chemistry, Jan Kochanowski University, 7 Uniwersytecka, 25-406 Kielce, Poland; Beata.Szczepanik@ujk.edu.pl; 2Institute of Environmental Engineering Polish Academy of Sciences, 34 M. Skłodowskiej-Curie St., 41-819 Zabrze, Poland; marianna.czaplicka@ipis.zabrze.pl

**Keywords:** inverse liquid chromatography, adsorption, phenol, chlorophenols, halloysite

## Abstract

The adsorption of phenol, 2-, 3-, 4-chlorophenol, 2-, 4-dichlorophenol and 2-, 4-, 6-trichloro-phenol on halloysite nanotubes modified with hexadecyltrimethylammonium bromide (HDTMA/halloysite nanocomposite) was investigated in this work by inverse liquid chromatography methods. Morphological and structural changes of the HDTMA/halloysite nanocomposite were characterized by scanning and transmission electron microscopy (SEM, TEM), Fourier-transform infrared spectrometry (FT-IR) and the low-temperature nitrogen adsorption method. Specific surface energy heterogeneity profiles and acid base properties of halloysite and HDTMA/halloysite nanocomposite have been determined with the inverse gas chromatography method. Inverse liquid chromatography methods: the Peak Division and the Breakthrough Curves Methods were used in adsorption experiments to determine adsorption parameters. The obtained experimental adsorption data were well represented by the Langmuir multi-center adsorption model.

## 1. Introduction

Phenols and its chloro derivatives are extensively used in numerous industrial processes [1,2]. Pharmaceutical, petrochemical, pesticides, plastic, textile, paper and other industries were reported as major sources of these compounds in wastewater [2]. Additionally, phenol and chlorophenol residues appear in the environment as a result of burning of urban waste, combustion of organic matters and biodegradation of pesticides [3]. Chloro derivatives of phenol show mutagenic and carcinogenic effect and low biological degradability even at trace levels.

That is why they are listed as priority pollutants according to the US Environmental Protection Agency (USEPA) and the U.S. Agency for Toxic Substances and Disease Registry [2,4]. The toxicity of chlorophenols is affected by the number and position of chlorine atoms in the benzene ring [3]. Their environmental sustainability, poor biodegradability and unpleasant odor cause the need to remove these compounds from the environment. Their removal from industrial effluents is currently one of the challenges in water purification and several treatment methods, including biological, chemical and physical processes are being applied for this purpose [5]. One of the effective processes for the removal of phenol and chlorophenols is adsorption due to its high efficiency, simplicity and applicability [6].

Numerous and various adsorbents have been studied for phenol and chlorophenols removal in the last few years. They can be classified as belonging to the following groups: carbonaceus adsorbents [1,2,7,8,9,10], polymeric adsorbents [10,11,12,13,14,15], zeolites [16,17,18], clays and other adsorbents [10]. Naturally abundant clays have received a lot of attention due to their low cost, chemical stability, high adsorption capacity and ion exchange properties [19]. Commonly used clays as adsorbents of phenolic compounds were monmorillonite [20,21,22,23,24] or bentonite [25,26,27]. Adsorption capacities of various adsorbents for phenol and its chloro derivatives are given in Table 1.

The modification of clays with surfactant increases their hydrophobicity and therefore enhances their adsorption capacity toward the organic pollutants [21,22,28,29,30]. Na-Montmorillonite modified with different surfactants was used to remove 2-naphthol, 2-, 4-chlorophenol, 2-, 4-, 6-trichlorophenol, 4-nitrophenol, aniline and bisphenol A in the adsorption process [22]. Kaolinite and halloysite modified with hexadecyltrimethylammonium bromide (HDTMA) were used as adsorbents of naphthalene [31,32].

Halloysite is an aluminosilicate dioctahedral 1:1 clay mineral (formula: Al_2_(OH)_4_Si_2_O_5_ × nH_2_O) with a tubular nanostructure (1D nanomaterial). The outer surface of halloysite nanotubes consists of siloxane groups (Si-O-Si) and is negatively charged, while the inner lumen consists of aluminol (Al-OH) groups and is positively charged. This allows one to modify the surface properties (charge, hydrophilicity) of halloysite by physically adsorbing some specific cations, for example cationic surfactants. Surfactant (for example quaternary ammonium cations) modification affects the change of halloysite surface, which becomes more hydrophobic. As a consequence, it can adsorb nonpolar organic solutes or anions [20,33,34,35,36,37,38].

In this work, the halloysite modified HDTMA was used as adsorbent of phenol and chlorophenols for the first time to our best knowledge. The Peak Division and the Breakthrough Curves Inverse Liquid Chromatography (PD ILC, BC ILC) methods [39] were applied to obtain adsorption constants and adsorbent maximum capacities for the adsorption process.

## 2. Materials and Methods

### 2.1. Chemicals and Materials

Halloysite was obtained from the “Dunino” strip mine, Intermark Company, Legnica, Poland. Phenol ≥ 99%, 2-, 3-, 4-chlorophenol ≥ 99%, 2-, 4-dichlorophenol ≥ 99% were purchased from Sigma-Aldrich, Poznań, Poland. Hexadecyltrimethylammonium bromide ([(C_16_H_33_)N(CH_3_)_3_]Br and 2-, 4-, 6-trichlorofenol ≥ 98% were acquired from MERCK, Darmstadt, Germany. Sodium chloride 95% was from Avantor Performance Materials Poland S.A, Gliwice, Poland. Deionized water (1.74 μs∙cm^−1^, temp. 25 °C) (Direct-Q UV Water Purification System, MerckMillipore, Burlington, Massachusetts, USA) was used through all experiments.

### 2.2. Preparation of HDTMA/Halloysite

Raw halloysite (HAL) was washed with deionized water dried at 60 °C for 24 h. Particle fraction of 0.4–0.63 mm was used for further preparation. The details of this preparation are described in Ref. [40]. Next, halloysite samples (100 g) were contacted with sodium chloride solution (1 mol/dm^3^, 24 h) to obtain Na-halloysite. The preparation of HDTMA/halloysite was conducted according to the following procedure: 100 cm^3^ of 0.002 M HDTMA solution was shaken with 1 g of Na-halloysite at ambient temperature for 24 h. Then, the obtained samples were decanted, washed with deionized water dried in air and labelled as HDTMA/HAL.

### 2.3. Characterization of Adsorbent

Textural properties of HDTMA/halloyite samples were characterized using low-temperature nitrogen adsorption-desorption isotherms method (−196 °C) on a volumetric adsorption analyzer ASAP 2020 by Micromeritics (Norcross, Georgia, GA, USA). Before measurements, the samples were degassed at a temperature of 200 °C for 2 h. Specific surface area (S_BET_) of the studied samples was determined with the Brunauer-Emmett-Teller (BET) method at relative pressure from 0.05 to 0.2, considering the surface occupied by a single molecule of nitrogen in an adsorptive monolayer (cross-sectional area equal 0.162 nm^2^) [41,42]. Total pore volume (Vt) (the sum of micropores volume (V_mi_) and mesopores (V_me_) was calculated from one point of nitrogen adsorption isotherm, corresponding to the relative pressure p/p_o_ equal 0.99 [42].

SEM image of halloysite was obtained using a Zeiss Ultra Plus Scanning Electron Microscope (Carl-Zeiss-Oberkochen, Jena, Germany) equipped with two secondary electron detectors (standard placed SE2 chamber and intra-column InLens) and two backscattered electrons detectors in the Nanostructures Laboratory (the High Pressure Institute of the Polish Academy of Sciences). STEM-HAADF (Scanning Transmission Electron Microscopy technique with the High Angle Annular Dark Field detector) was performed using the 200 kV FEI TecnaiOsiris transmission electron microscope (Electron Microscopy Laboratory, the Department of Chemistry, Jagiellonian University, Cracow, Poland) (FEI Company, Hillsboro, Oregon, USA), with samples loaded onto a lacey carbon coated copper grid (AGAR Scientific, Stansted, UK).

Infrared spectra were recorded using a Perkin-Elmer Spectrum 400 FT-IR/FT-NIR spectrometer (Perkin-Elmer Waltham, Massasuchetts, USA)with a smart endurance single bounce diamond, attenuated total reflection (ATR) cell. Spectra in the 4000–650 cm^−1^ range were obtained by the co-addition of 500 scans with a resolution of 4 cm^−1^. Before the measurements, all samples were dried and powdered in an agate mortar.

### 2.4. Inverse Liquid Chromatography Methods

#### 2.4.1. Adsorption Measurements by the PD ILC Method

Adsorption isotherms are determined with the peak profile, which is also named as peak division [43]. This method used in inverse liquid chromatography was presented in the work [44]. The PD ILC method requires dividing area bounded by adsorbate injection and the diffuse side of the chromatogram (also known as total adsorption surface) into *L* parallel parts to the chromatogram baseline; furthermore, it is essential to measure area of each segment [44]. Adsorption value *a* of the substance *i* can be calculated with the equation given below:(1)$$a=\frac{n\sum_{1}^{L}S_{Ls}}{mS_{p}}$$

*a*—the quantity of the adsorbed substance *i* on adsorbent (mg/mg)

*m*—adsorbent mass (mg)

*n*—mass of the substance *i* applied on the adsorbent (mg)

*S_p_*—adsorbate peak area (mV min)

*S_Ls_*—area of *L* part of area bounded by adsorbate injection and the diffuse side of the chromatogram (mV min)

Equilibrium concentration of substance *i* in the liquid phase, which corresponds to adsorption values $a_{i}$ of the substance *i*, is calculated by dividing chromatographic peak height into *i* parts according to the following equation:(2)$$c=\frac{n\sum_{1}^{I}h_{I}}{FS_{p}}$$
where:

*c*—equilibrium concentration of substance *i* in liquid phase (mg/cm^3^)

$h_{I}$—height of the peak part *i* (mV)

*F*—liquid phase flow (cm^3^/min)

The method described in Ref. [44] does not take into account the correction of the tailing edge of a chromatographic peak for diffusion and flow of the non-absorbing substance in this method of analyzing the elution peaks.

This article describes the development of the PD ILC method. This method is developed by additional measurements on the column filled, not adsorb substance, used as the reference system for column with adsorbent. The PD ILC method enables this correction, because it is possible to subtract definite parameters of the peaks.

The area bounded by adsorbate injection and the diffuse side of the chromatogram were calculated by equations (Figure 1) for column with adsorbent:(3)$$S_{s}=\sum_{1}^{L}S_{Ls}$$

And for reference column:(4)$$S_{s}^{r}=\sum_{1}^{K}S_{Ks}^{r}$$
where index *r* means reference column.

After inserting Equations (3) and (4) into Equation (1), Equation (5) is obtained:(5)$$a=\frac{n\left(\sum_{1}^{L}S_{Ls}-\sum_{1}^{K}S_{Ks}^{r}\right)}{m\left(S_{p}-S_{p}^{r}\right)}$$

The height of the chromatographic peaks was similarly calculated for column with adsorbent:(6)$$h=\sum_{1}^{l}h_{l}$$

And for reference column:(7)$$h^{r}=\sum_{1}^{k}h_{k}$$

After inserting Equations (6) and (7) into Equation (2), Equation (8) is obtained:(8)$$c=\frac{n\left(\sum_{1}^{k}h_{k}+\sum_{1}^{l}h_{l}\right)}{2F\left(S_{p}-S_{p}^{r}\right)}$$

Equation (8) averages values of height the peaks obtained from measurements of column with adsorbent and reference column.

Calculations concerning concentration $c_{i}$ of substance *i* in the liquid phase as well as the amount of the adsorbed adsorbate $a_{i}$ were conducted on the basis of the data concerning peak profile division contained in CDPS (Computer Data Peak Software) database [45]. As a result, relation *a = f(c)* is a function describing an adsorption isotherm. The number of points determining an isotherm depends on the selected number of area division into *L* and *K* segments.

The peak division method enables determining adsorption isotherms on the basis of a single chromatographic peak. The manner of conducting calculations was described in [39].

$S_{s}^{r}$, $S_{s}$—area bounded by adsorbate injection and the diffuse side of the chromatogram (area A, B, C, D) for reference column (1) and adsorbent column (3).

$S_{Ks}^{r}$, $S_{Ls}$—segment area corresponding to part *K*, *L* of total area A, B, C, D for reference column (21) and adsorbent Equation (3).

The division of the total height, *h*, of the adsorption peak into segments, where *k*, *l* is the height of the corresponding segment of the total height of chromatographic peak for reference column (2) and adsorbent column (4).

#### 2.4.2. Adsorption Measurements by the BC ILC Method

Adsorption measurements with the BC ILC method require using the liquid chromatograph equipped with precision pumps to ensure steady flow of liquid phase for a specified period of time, six-port valve systems to switch the chromatographic column on and off to regulate liquid phase flow and a high-sensitivity detector to measure concentration to determine breakthrough curves of an adsorbent. In this manner, continuous registration of concentration changes at the column outlet during adsorbate flow is possible. This concentration dependency changes as a function of time. It is described with a curve in the sigmoidal form and the change in concentration as a function of time is determined by the adsorption volume based on the parameters of this curve. The above-described measurement ILC method has limitations in the case when adsorbate slowly eluates from the column. Then, the sigmoidal curve strongly tilts with respect to the time axis. It differs significantly from the shape of a rectangular pulse. End timing of adsorbate concentration measurement can be incorrect.

A new modified method of measurement determining the adsorption volume with the BC ILC method has been developed. This modification includes correction consisting of the calculation of the area under the sigmoidal curve. The sigmoidal curve is created from the product of difference between the end time of measuring adsorbate concentration at the outlet of the column and start time of adsorbate concentration measurement at the column outlet and detector signal. The concentration of adsorbate versus time in the outlet on the chromatographic column is calculated from the product of difference between the end time and the start time of measuring adsorbate concentration at the outlet of the column and signal of the detector (Figure 2). The difference between the values of these two areas can calculate the mass of the adsorbed substances in the column. This modification is aimed at the column with adsorbent with respect to the comparative column (Figure 2A,B).

The mass of the adsorbed substances in the column with the adsorbent and in the reference column was calculated according to the following formula:(9)$$a=\frac{c\times F\left[\frac{S_{a}-S_{a}^{p})}{h_{b}}-\frac{(S_{s}-S_{s}^{p})\;}{h_{wz}}\right]}{m}$$

*a*—the number of adsorbate milligrams adsorbed per unit mass of adsorbent (mg g^−1^)

*c*—adsorbate concentration (mg cm^−3^)

*F*—liquid phase flow (cm^3^ min^−1^)

*m*—adsorbent mass (g)

$t_{0}^{s}$—start time of adsorbate concentration measurement for the reference column (min)

$t_{e}^{s}$—end time of adsorbate concentration measurement for the reference column (min)

$t_{p}^{s}$—start time of adsorbate concentration measurement at the reference column outlet (min)

$h_{s}$—signal value for the reference column (min)

$S_{s}$—area; $S_{s}=h_{s}\times \left(t_{e}^{s}-t_{0}^{s}\right)$ for the reference column (mV min)

$S_{s}^{p}=\int_{t_{p}^{s}}^{t_{e}^{s}}f\left(h\right)dt$—field corresponding to variable adsorbate concentration at the reference column outlet (mV min)

$t_{0}^{a}$—start time of adsorbate concentration measurement for the column with adsorbent (min)

$t_{e}^{a}$—end time of adsorbate concentration measurement for the column with adsorbent (min)

$t_{p}^{a}$—start time of adsorbate concentration measurement at the outlet of column with adsorbent (min)

$h_{a}$—signal value for the column with adsorbent (min)

$S_{a}$—area; $S_{a}=h_{a}\times \left(t_{e}^{a}-t_{0}^{a}\right)$ for the column with adsorbent (mV min)

$S_{a}^{p}=\int_{t_{p}^{a}}^{t_{e}^{a}}f\left(h\right)dt$—field corresponding to the variable adsorbate concentration at the column outlet with adsorbent (mV min)

### 2.5. Surface Characteristics of the Adsorbents by Inverse GAS Chromatography Method

Surface characteristics of the adsorbents can also be successfully determined with the use of the inverse gas chromatographic (IGC) technique. The technique is a useful method to evaluate adsorption isotherms, dispersion and specific interfacial energy profiles of adsorbent at low partial pressures of adsorbates.

The Schultz method also utilizes adhesion work equation $\left(W_{a}\right)$, together with free adsorption energy of the methylene group $\Delta G^{CH_{2}}$, which is analogical to the Dorris-Gray method [46,47] in describing mutual interaction of the examined substance (adsorbent) with the test one $\Delta G^{CH_{2}}$
(10)$$W_{a}=W_{a}^{d}=2\sqrt{\gamma_{s}{}^{d}\times \gamma_{l}{}^{d}}$$
where:

$\gamma_{s}^{d}$—dispersive component of surface energy [mJ m^−2^]

$\gamma_{l}^{d}$—dispersive energy of probe molecule [mJ/m^−2^]

$W_{a}^{d}$—dispersive component of work of adhesion [mJ]
(11)$$-\Delta G^{CH_{2}}=N_{A}\times a_{CH_{2}}\times W_{a}$$

$N_{A}$—Avogadro number

Cross-section area $a_{CH_{2}}$ of the alkane molecule is provided with the following equation:(12)aCH2=1.09×1014×(Mρ·NA)23

*M*—molar mass [g]

ρ—density [cm^3^ g^−1^]

The transition of the three equations enables obtaining an expression in the linear form, from which $\gamma_{s}{}^{d}$ can be determined.
(13)$$RT\cdot ln\left(V_{N,n}^{’}\right)=2N_{A}\times a_{CH_{2}}\times \sqrt{\gamma_{s}{}^{d}}\times \sqrt{\gamma_{l}{}^{d}}+C$$

*C*—constant value

The dependence diagram $RT\cdot ln\left(V_{N,n}^{’}\right)=f\left(a_{CH_{2}}\times \sqrt{\gamma_{l}{}^{d}}\right)$ is a straight line. The slope coefficient facilitates determining the value of dispersive free surface energy at the stationary phase. Polar substances that are to be dosed into the column with the test substance in identical conditions of the conducted measurements are not present in the n-alkane line. Vertical distance on the axis $RT\cdot ln\left(V_{N,n}^{’}\right)\;$ from the n-alkane line to the point of polar substance is a component of specific adsorption energy $\Delta G_{a}^{sp}$ [48]. Therefore, the Schultz method enables determining $\gamma_{s}{}^{d}$ and $\Delta G_{a}^{sp}$.

IGC allows testing acidic-basic properties of solids. For this reason, polar substances with known donor-acceptor properties are applied. They can be acidic (electron acceptors), basic (electron donors) or amphoteric in character. A modified Gutmann equation [49] is used for the following calculations:(14)$$\Delta G_{a}^{sp}=DN\times K_{a}+AN\times K_{b}$$

$K_{a}$—acidic characteristic of solid

$K_{b}$—basic characteristic of solid

AN—acceptor number

DN—donor number
(15)$$\frac{\Delta G_{a}^{sp}}{AN}=\frac{DN}{AN}\times K_{a}+K_{b}$$

While creating the dependence diagram $\frac{\Delta G_{a}^{sp}}{AN}=f\left(\frac{DN}{AN}\right)$, constant $\;K_{a}$ can be calculated from the slope coefficient of the diagram; in turn, the value $K_{b}$ can be calculated using OY axis intersection. Ratio $\frac{K_{b}}{\;K_{a}}$ facilitates specifying the character of the test surface. If the ratio is $\frac{K_{b}}{\;K_{a}}>1$, then the surface is basic (donor properties prevail over the acceptor ones). If, however, the ratio is $\frac{K_{b}}{\;K_{a}}<1$, then the surface is acidic. On the other hand, if $\frac{K_{b}}{\;K_{a}}\approx 1$, then the surface is amphotheric [50].

### 2.6. Chromatographic Measurements

Thermo Scientific Dionex UltiMate 3000 Series chromatography system (Thermo Fisher Scientific Inc., Waltham, MA, USA) was used in chromatographic experiments with a diode array UV detector (DAD 190-800 nm) and Chromeleon software. The DAD system was equipped with an additional exit of the analog signal to which the analog-to-digital converter was connected. The signal from this converter was directed to the second computer with the KSPD software (Metroster, Toruń, Poland).

Apparatus scheme used for PD ILC and BC ILC methods were described elsewhere [39]. Adsorption measurements (PD ILC method) for phenol and its chloro derivatives were conducted with the chromatographic column, 10 cm in length and with internal diameter of 0.8 mm. The column contained ca. 1 g of halloysite adsorbent with grain diameter of 0.4–0.63 mm. It had been conditioned prior to measurement, i.e. with redistilled water (1.74 µS/cm), flow rate of 0.5 mL/min and at pressure value of ca. 42 bar. An analogical column with the same adsorbent, i.e. a comparative column, was used as a reference system. A comparative column was conditioned in the same manner. Next, adsorbate (20–50 µL with concentration of 250–500 mg/dm^3^) was applied to both columns. If the surfaces of adsorption peaks (obtained on both columns) did not differ by more than 6%, the measurement result was considered correct. Adsorption measurements were conducted at a temperature range of 298–313 K.

Adsorption measurements (BC ILC method) were carried out using a column about the same size as described above. The column was filled with 1 g of halloysite adsorbent with a diameter of granules of 0.4–0.63 mm. The column had been conditioned for 1 h prior to measurement using redistilled water (1.74 µs/cm), with flow rate 0.5 mL/min and pressure at about 42 bar. A solution of an adsorbate with the concentration in the range 5–60 mg/dm^3^ was dosed to the prepared column. As the reference, a column with the same size was used, filled with silanized glass beads. Measurements of adsorption were conducted at a temperature range of 298–313 K.

Both adsorption measurement (PD ILC and BC ILC) of adsorbate concentration was performed by applying wavelength UV-DAD detector phenol 270 nm; 2-chlorophenol 274 nm; 3-chlorophenol 274 nm; 4-chlorophenol 280 nm; 2-, 4-dichlorophenol 284 nm; 2-, 4-, 6-trichlorophenol 294 nm, respectively.

Specific surface energy heterogeneity profiles and acid base properties of HAL and HDTMA/HAL adsorbents were determined with an inverse gas chromatograph, with the Surface Energy Analyzer (IGC-SEA) of Surface Measurement Systems Ltd. (Alperton, UK) at low surface coverage. Experiments were conducted at 423 K with a helium carrier gas flow rate of 20 cm^3^/min using a flame-ionization detector, methane gas as marker for the hold-up time and n-alkanes, methanol, acetononitrile together with ethyl acetate as probe compounds.

## 3. Results and Discussion

### Characterization of the HDTMA/Halloysite Adsorbent

Nitrogen adsorption-desorption isotherms for HAL and HDTMA/HAL samples are characteristic as regards type IV with H3 hysteresis loops, according to the International Union of Pure and Applied Chemistry (IUPAC) classification (Figure 3) [50,51]. It proved that the examined samples belong to the group of mesoporous materials, which is confirmed by data given in Table 2.

Comparison of parameters from Table 2 shows that after modification with the HDTMA cationic surfactant, adsorbent specific surface area (S_BET_) decreased from 47 m^2^ g^−1^ to 43 m^2^∙g^−1^, total pore volume (V_t_) decreased from 0.1773 cm^3^∙g^−1^ to 0.1716 cm^3^∙g^−1^ and pore diameter increased slightly (from 16.8 nm to 17.6 nm). The decrease in the porous structure parameters may result from partial blocking of the halloysite surface by HDTMA particles used for mineral modification.

ATR FT-IR spectra of HAL and HDTMA/HAL samples in the 4000–650 cm^−1^ region are presented in Figure 4. HAL spectrum sample shows characteristic bands for the kaolin-group minerals: in the 3700–3600 cm^−1^ region, the vibration of the OH group and in the 1750–650 cm^−1^ region bands assigned to Si-O (1107 cm^−1^) as well as to perpendicular stretching vibrations of Si-O-Si (1030 and 691 cm^−1^) [52]. The two new bands at 2924 and 2852 cm^−1^ appear in the HDTMA/HAL sample spectrum (Figure 3), which can be assigned to the CH_2_ stretching vibrational bands in the alkyl chain of HDTMA and are almost the same bands as in HDTMA micelles [31]. This fact suggests that micelle-like clusters or double layer of HDTMA chains may form on the halloysite surface, rather than individual molecules distributing themselves uniformly over the halloysite surface [31,51].

SEM and TEM images (Figure 5) confirm the tubular structure of the halloysite (HAL) sample. Modification of halloysite with HDTMA does not change the mineral structure significantly.

Chemical composition of halloysite obtained by the Wavelength Dispersive X-ray Fluorescence (WDXRF) analysis [40] is as follows: Al_2_O_3_ 42.3%, SiO_2_ 50.7%, Fe_2_O_3_ 2.99%, TiO_2_ 1.41%, CaO 0.29%, MgO 0.06%, Na_2_O 0.12%, K_2_O 0.04%, P_2_O_5_ 0.51%, SO_3_ 0.17%. In the diffractograms of the halloysite sample (XRD method), the peaks of the following minerals were identified: halloysite, kaolinite, hematite, calcite [40].

Surface properties of HAL and HDTMA/HAL samples are presented in Table 3. The values of dispersive free adsorption energy of methanol, acetononitrile and ethylacetate adsorbates decreased for HDTMA/HAL compared with these values for HAL. The ratio of $\frac{K_{b}}{\;K_{a}}$ is < 1 for HAL adsorbent pointing the acidic character of adsorbent surface. Presence of HDTMA cations on the adsorbent surface changes the character of the surface; the ratio $\frac{K_{b}}{\;K_{a}}$ is equal 1.06, so the surface becomes basic (donor properties prevail over the acceptor ones) or amphoteric ($\frac{K_{b}}{\;K_{a}}\approx 1)$.

Adsorption measurements carried out for phenol and chlorophenols on HAL and HDTMA/HAL adsorbents showed that unmodified halloysite did not practically adsorb these compounds. Therefore, further studies were performed for the HDTMA/HAL adsorbent.

Most often, the linear [24,27,31] Langmuir [24,25,31], Freundlich [25,27,31], Temkin and Dubinin–Radushkevich [27] equations were adjusted to the results of adsorption measurements in the literature relating to the adsorption process on halloysite adsorbents. Adsorption of phenol and its chloro derivatives from aqueous solutions on various adsorbents such as montmorillonite, activated carbon, waste materials from industries, agricultural by-products and biomass-based activated carbon [2,10,20,21,53] occurred according to Langmuir or Freundlich models. It is assumed that the isothermal adsorption of phenol and its chloro derivatives on halloysite adsorbent HDTMA *a_i_* = *f*(*c*_i_) can also correspond to these adsorption models. The results showed the best correlation of experimental data for the Langmuir model of adsorption. The following cases in the Langmuir model were taken into consideration: adsorption on one active center without dissociation, adsorption on two active centers without dissociation, adsorption on multiple active centers without dissociation and adsorption on two active centers with dissociation. The selected adsorption models and the corresponding modified forms of the Langmuir and Freundlich equations (Equations (16)–(19)) are presented in Table 4 [54,55]. It was assumed that, apart from an exactly defined mechanism for the adsorption processes of phenol (as well as its chloro derivatives), e.g.single- and two-center mechanisms, more complex adsorption mechanisms can also be present. Therefore, an equation describing a multicenter adsorption model (e.g., Equations (17) and (18) Table 4) can be applied to determine the number of centers. Curve calculations and simulations were completed with the Levenberg–Marquardt least-squares method using the Origin Microcal Program Origin User’s Manual; Microcal Software Inc.: Northampton, MA, USA, 2018 [56].

Figure 6 shows the curves obtained from the application of Langmuir and Freundlich Equations (16)–(19) with the least-squares method to the adsorption data of phenol, 2-chlorophenol, 3-chlorophenol, 4-chlorophenol, 2-, 4-dichlorophenol and 2-, 4-, 6-trichlorophenol on HDTMA/HAL adsorbent at 298 K. Adsorption isotherms data for adsorbates at 298 K on adsorbent HDTMA/HAL were shown in Appendix A
Appendix A.

As can be seen from the plot, only the curves determined by Equations (17)–(19) coincided with the experimental points. The remaining curve, determined by Equation (16), did not overlap with the experimental points. The values of *χ^2^* and *R^2^* in Table 5 confirm the best fit of Equation (17) to the experimental data. The adsorption isotherm of phenol determined with Equation (18) was not on the scale (Figure 6). Designated equilibrium constants for the adsorption did not have a physical meaning and therefore this equation was omitted in the calculations.

The values for the equilibrium constants for phenol, 2-chlorophenol, 3-chlorophenol, 4-chlorophenol, 2-, 4-dichlorophenol and 2-, 4-, 6-trichlorophenol on HDTMA/HAL, calculated on the basis of Equation (17) within the temperature range of 298–313 K, are given in Table 6. Adsorption enthalpy was calculated based on the Van’t Hoff equation. The determined enthalpy value of adsorption and the related adsorption entropy can provide correctness indication of the measurements if they conform to Boudart’s rules [57].

The value of adsorption entropy must be negative:(20)$$\Delta S_{a}^{0}<0$$

Because molecule entropy decreases as the number of freedom degrees diminishes during the course of the adsorption process.

The absolute value of adsorption entropy calculated under standard conditions must be smaller than the standard entropy value of the molecule formation:(21)$$\left|\Delta S_{a}^{0}\right|<S_{298}^{0}$$

This is because molecule adsorption cannot lead to a decrease in its entropy greater than the value of its absolute entropy of formation.

As can be seen from the data illustrated in Table 7, the determined adsorption enthalpies agree with Boudart’s rules and the determined equilibrium constants for phenol, 2-chlorophenol, 3-chlorophenol, 4-chlorophenol, 2-, 4-dichlorophenol and 2-, 4-, 6-trichlorophenol on HDTMA/HAL exhibit thermodynamic correctness.

Adsorption enthalpy for phenol equals −11.6 kJ∙mol^−1^, for 2-chlorophenol −23.3 kJ∙mol^−1^, for 3-chlorophenol −26.3 kJ∙mol^−1^, for 4-chlorophenol −22.9 kJ∙mol^−1^, for 2-, 4-dichlorophenol −26.1 kJ∙mol^−1^ and for 2-, 4-, 6-trichlorophenol −18.4 kJ∙mol^−1^.

In Equation (17), describing the multi-center adsorption, the value of *n* exponent determines the number of active centers participating in the phenol and its chloro derivatives adsorption process on HDTMA/HAL. The determined values are not whole numbers, which for phenol equals 1.09. The number of active centers (localized sites on adsorbent surface) participating in the adsorption process of 2-chlorophenol on HDTMA/HAL is 1.25. A similar situation can be observed in the case of 3-chlorophenol, i.e., the value equals 1.39. In addition, in the adsorption process of 4-chlorophenol, the number of active centers is 1.47, for 2-, 4–dichlorophenol it equals 1.19 and for 2-, 4-, 6–trichlorophenol it is 1.42.

As can be seen from Table 8, adsorption capacity decreases towards phenol. 2-chlorophenol, 4-chlorophenol, 3-chlorophenol, 2-, 4-dichlorophenol and 2-, 4-, 6-trichlorophenol on HDTMA/HAL adsorbent.

## 4. Conclusions

FTIR spectra and inverse gas chromatographic measurements confirmed that the modification of halloysite nanotubes by HDTMA changes the character of halloysite surface, enabling the adsorption of phenol and chlorophenols from negative to positive charged and more hydrophobic. The adsorption of phenol, 2-, 3-, 4-chlorophenol, 2-, 4-dichlorophenol and 2-, 4-, 6-trichlorophenol on HDTMA/halloysite nanocomposite was studied with the use of inverse liquid chromatography methods: the Peak Division and the Breakthrough Curves methods. The obtained experimental adsorption data were well represented with the Langmuir (multi-center) adsorption model (Langmuir adsorption model on multiple active centers without dissociation). Adsorption constant values decreased for phenol and monochlorophenols in the following order: phenol > 4-chlorophenol > 3-chlorophenol and for others chloro derivatives: 2-, 4-dichlorophenol > 2-, 4-, 6-trichlorophenol. The values of factor *n* were fractional, indicating the adsorption mechanisms of phenol and chlorophenols with a different number of adsorptive centers on the HDTMA/halloysite surface. The obtained results confirm that HDTMA/halloysite materials are suitable adsorbents for phenol and chlorophenols.

Phenol and chlorophenols adsorption measurements were carried out in the flow system, mainly on carbon adsorbents and resins [58,59,60,61]. In these works, the breakthrough curve method was used to determine the adsorption capacity of adsorbent.

According to our knowledge, we applied the Peak Division and the Breakthrough Curves ILC methods for the adsorption phenol and chlorophenols on halloysite modified by HDTMA. The first method can be used to determine the form of the adsorption equation and the second one can be used to determine the adsorption capacity of the adsorbent. In addition, these methods provides experimental simplicity combined with low adsorbate and solvent consumption.

## Figures and Tables

**Figure 1 materials-13-03309-f001:**
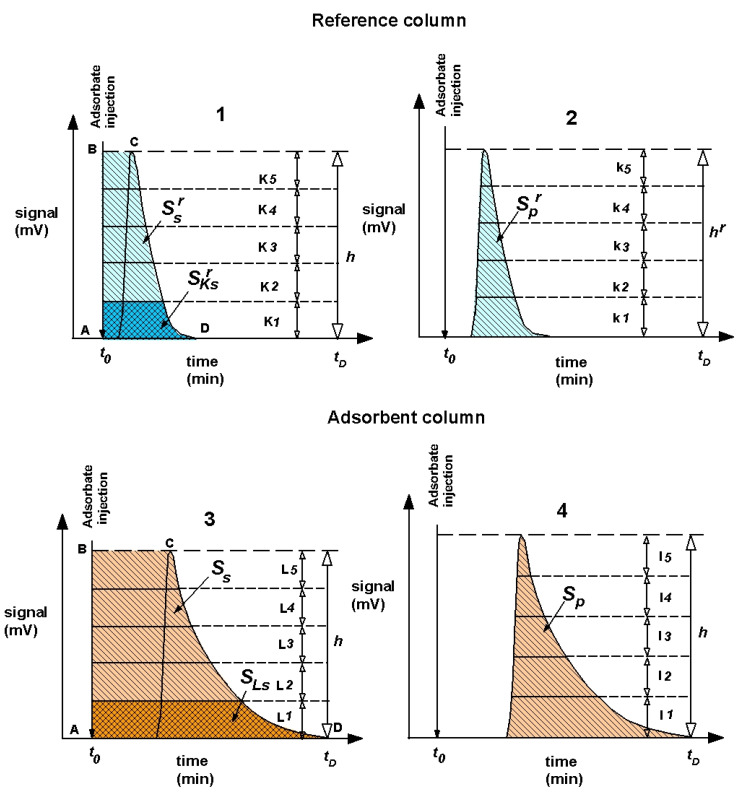
The illustration of applying the peak division method to calculate the value for drawing the isotherm.

**Figure 2 materials-13-03309-f002:**
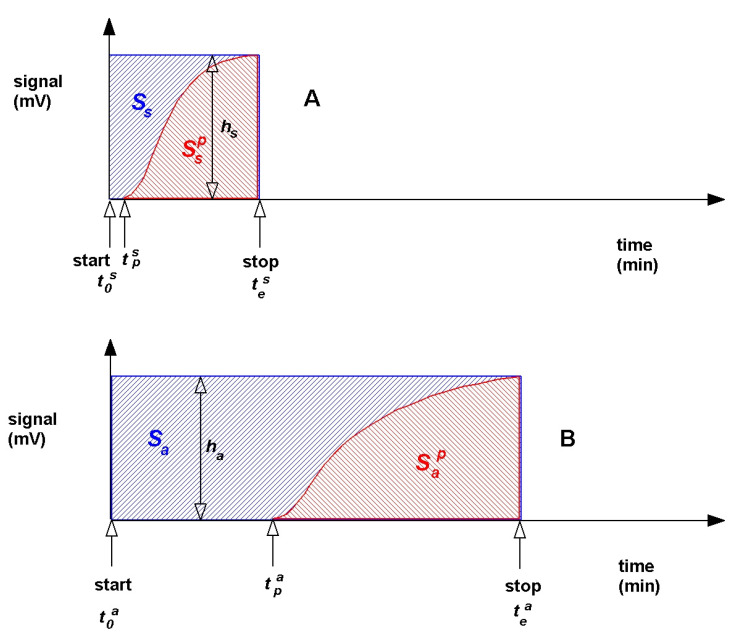
The concentration of adsorbate versus time in the outlet on the chromatographic column: (**A**) reference column and (**B**) adsorbent column.

**Figure 3 materials-13-03309-f003:**
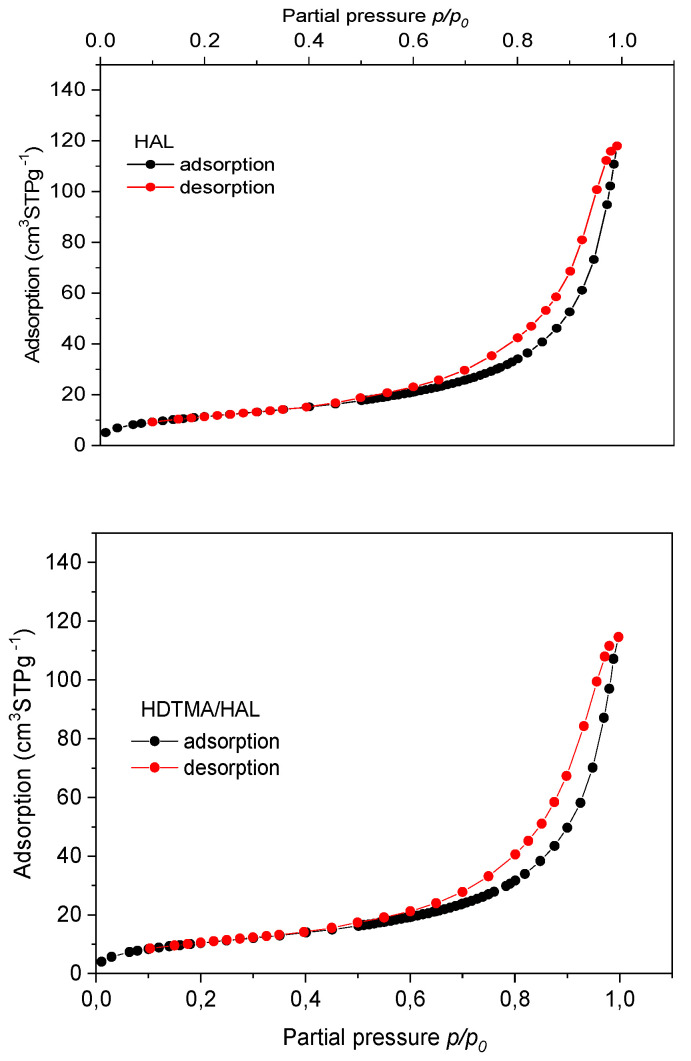
Adsorption and desorption isotherm for nitrogen for HAL and HDTMA/HAL adsorbents.

**Figure 4 materials-13-03309-f004:**
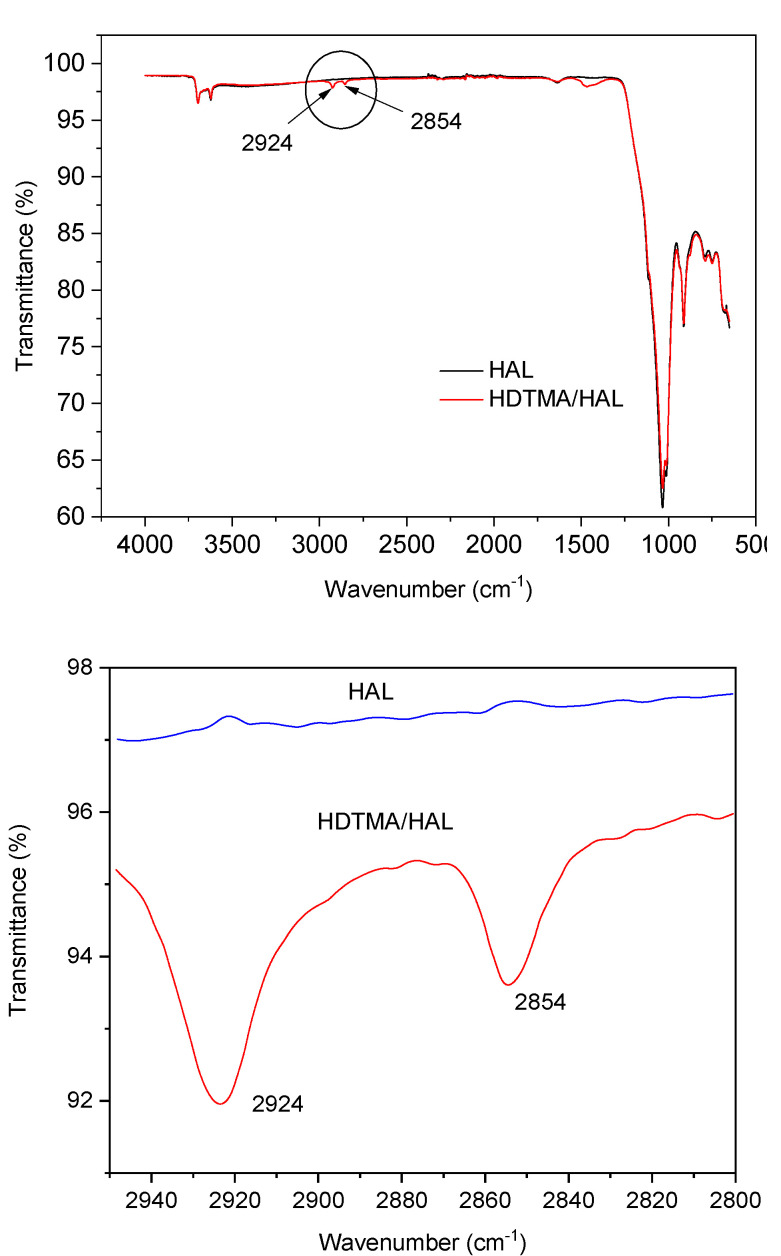
FTIR spectra of Hal and HDTMA/HAL samples.

**Figure 5 materials-13-03309-f005:**
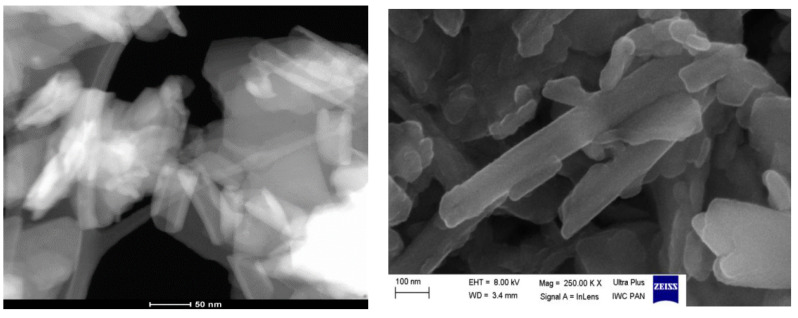
(**left**) TEM and (**right**) SEM images of the HAL sample.

**Figure 6 materials-13-03309-f006:**
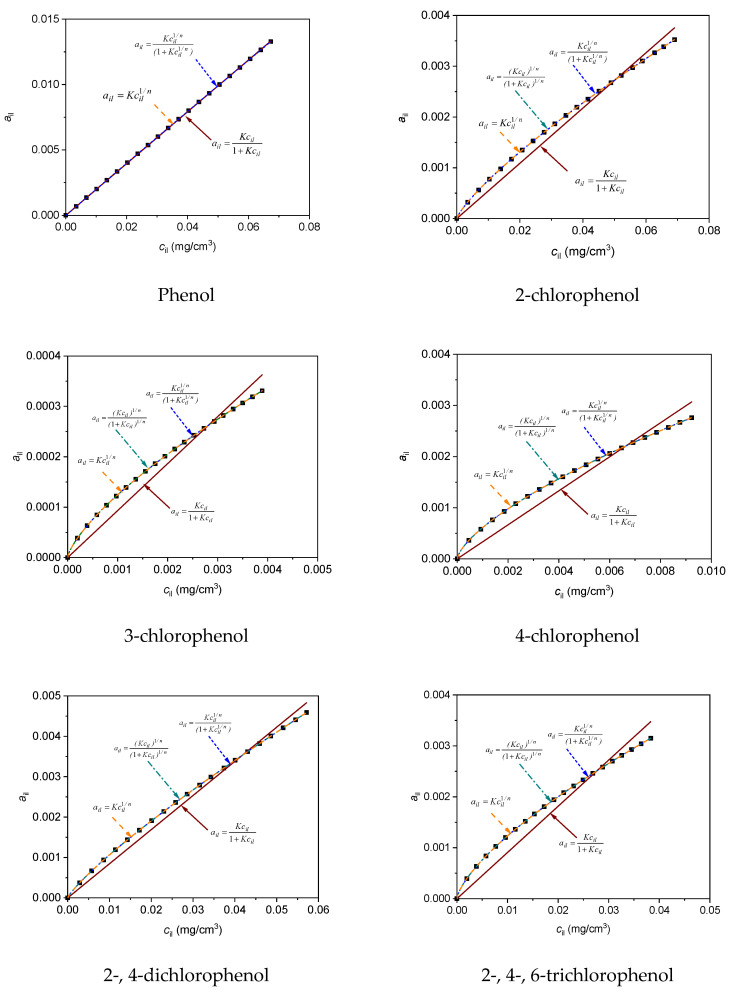
Adsorption isotherms for adsorbates at 298 K on adsorbent HDTMA/HAL. Solid lines represent curves obtained from the application of Langmuir Equations (16)–(19) to the adsorption data as adjusted by the least-squares method.

**Table 1 materials-13-03309-t001:** Adsorption capacity of various adsorbents for phenol and its chloro derivatives [1].

Adsorbate	Adsorbent	Adsorption Capacity (mg/g)
Phenol	Activated Carbons (Granular, Commercial, Powdered, Activated)	350.0, 322.5, 303.0, 283.3
	Sawdust	64.0
	Chitosan	59.74
	Red Mud	59.2
	Hollow Mesoporous Carbon Sphere	207.8
	Magnetic Nanoparticles	123.45
	Graphene Aerogels-Mesoporous Silica	90.0
	Multiwalled Carbon Nanotubes	64.6
	HDTMA-Clinoptilolite Zeolite	11.4 [10]
	HDTMA-Montmorillonite	94.9 [10]
2-Chlorophenol	Red Mud	117.3
	Carbon Nanotubes	86.1
	Chitosan	70.52
	Hypersol-Macronet Resins	125.7, 136.2 [10]
	Natural Clay	23.59 [2]
4-Chlorophenol	Activated Carbons (Commercial, Granular, Activated)	500.0, 319.9, 280.0
	Red Mud	127.1
	Chitosan	96.43
	Carbon Nanotubes	51.5
	Hypersol-Macronet Resins	144.4, 163.2
	Amberlite XAD-16	291.6 [10]
2-, 4-Dichlorophenol	Corn Cob Activated Carbon	451.2
	Chitosan	315.46
	Red Mud	130.0
2-, 5-Dichlorophenol	Porous Clay Heterostructure	45.5
3-, 4-Dichlorophenol	Porous Clay Heterostructure	48.7
2-, 4-, 6-Trichlorophenol	Coconut Husk Activated Carbon	716.1
	Chitosan	375.94
	Coconut Shell Activated Carbon	122.34
	Copper (II)-Halloysite Nanotubes	135.06
	Na-Montmorillonite Modified Different Surfactants	328.9, 306.7 [24]

**Table 2 materials-13-03309-t002:** Surfaces properties of adsorbents determined from isotherms of nitrogen.

Parameter	HAL	HDTMA/HAL
Surface Area S_BET_ (m^2^/g)	47	43
Total Pore Volume V_t_ (cm^3^/g)	0.1773	0.1716
Micropore Volume V_mik_ (cm^3^/g)	0.0061	0.0058
Mezopore Volume V_mez_ (cm^3^/g)	0.1713	0.1658
Pore Diameter (nm)	16.8	17.6

**Table 3 materials-13-03309-t003:** Surface properties of the HAL and HDTMA/HAL samples.

Adsorbent	Dispersive Free Adsorption Energy of Adsorbate (kJ/mol)			*K_B_*	*K_a_*	*K_b_/K_a_*
	Methanol	Acetonitryle	Ethylacetate			
HAL	−7.3	−13.1	−11.7	0.41	0.64	0.64
HDTMA/HAL	−4.5	−9.3	−8.5	0.46	0.43	1.06

**Table 4 materials-13-03309-t004:** The proposed adsorption models and forms of Langmuir and Freundlich equations.

Number of Equation	Equation	Adsorption Model
(16) ^a^	$a_{i\;}=\frac{Kc_{i}}{1\;+\;Kc_{i}}$	one-center adsorption without dissociation
(17) ^b^	$a_{i\;}=\frac{Kc_{i}^{1/n}}{\left(1\;+\;Kc_{i}^{1/n}\right)}$	*n*-center adsorption without dissociation
(18) ^b^	$a_{i\;}=\frac{{\left(Kc_{i}\right)}^{1/n}}{{\left(1\;+\;Kc_{i}\right)}^{1/n}}$	*n*-center adsorption with dissociation
(19) ^b^	$a_{i\;}=Kc_{i}^{1/n}$	Freundlich adsorption

^a^*K*—adsorption equilibrium constant (cm^3^/mg); ^b^*K*—adsorption equilibrium constant (cm^3^/mg)^1/*n*^.

**Table 5 materials-13-03309-t005:** Fit of adsorption equilibrium equation to the experimental data by the Levenberg-Marquardt least-squares method; analysis of variance.

Number of Equation	(16)	(17)	(18)	(19)
Phenol				
Adsorption Equilibrium Constant K	0.098	0.201	-	0.193
Error	0.0586	0.06529	-	0.0263
Coefficient *n*		1.09	-	0.99
Error	-	0.0043	-	0.0458
Chi-Square Minimization	4.46 × 10^−8^	3.34 × 10^−10^	-	2.90 × 10^−9^
Regression Coefficient	0.9745	0.9999	-	0.9879
2-chlorophenol				
Adsorption Equilibrium Constant K	0.027	0.030	0.040	0.025
Error	0.0043	0.0041	0.0231	0.0741
Coefficient *n*	-	1.25	1.18	1.25
Error	-	0.0087	0.034	0.0928
Chi-Square Minimization	5.39 × 10^−7^	2.35 × 10^−10^	1.46 × 10^−8^	1.29 × 10^−8^
Regression Coefficient	0.9762	0.9999	0.9235	0.9829
3-chlorophenol				
Adsorption Equilibrium Constant K	0.046	0.018	0.004	0.017
Error	0.0003	0.0009	0.0034	0.0819
Coefficient *n*	-	1.39	1.39	1.39
Error	-	0.0007	0.098	0.0824
Chi-Square Minimization	5.44 × 10^−8^	4.77 × 10^−9^	9.16 × 10^−7^	5.28 × 10^−6^
Regression Coefficient	0.9463	0.9998	0.9526	0.9833
4-chlorophenol				
Adsorption Equilibrium Constant K	0.016	0.067	0.019	0.066
Error	0.0043	0.0024	0.0086	0.0342
Coefficient *n*	-	1.47	1.47	1.47
Error	-	0.0011	0.068	0.0674
Chi-Square Minimization	4.90 × 10^−8^	3.21 × 10^−10^	8.19 × 10^−7^	6.78 × 10^−8^
Regression Coefficient	0.9233	0.9997	0.9832	0.9752
2-, 4-dichlorophenol				
Adsorption Equilibrium Constant K	0.042	0.051	0.028	0.050
Error	0.0054	0.0015	0.0018	0.0280
Coefficient *n*	-	1.19	1.20	1.20
Error	-	0.0017	0.0821	0.0215
Chi-Square Minimization	2.47 × 10^−8^	1.15 × 10^−10^	1.87 × 10^−6^	5.18 × 10^−8^
Regression Coefficient	0.9871	0.9989	0.9643	0.9869
2-, 4-, 6-trichlorophenol				
Adsorption Equilibrium Constant K	0.045	0.031	0.006	0.031
Error	0.0014	0.0023	0.0054	0.0305
Coefficient *n*	-	1.42	1.45	1.42
Error	-	0.0063	0.0653	0.0116
Chi-Square Minimization	5.34 × 10^−8^	1.18 × 10^−10^	1.07 × 10^−7^	3.67 × 10^−8^
Regression Coefficient	0.9352	0.9975	0.9876	0.9874

**Table 6 materials-13-03309-t006:** Adsorption equilibrium constants for phenol (PH), 2-chlorophenol (2CPH), 3-chlorophenol (3PH), 4-chlorophenol (4CPH), 2-, 4-dichlorophenol (24DCPH) and 2-, 4-, 6-trichlorophenol (246TCPH) on HDTMA/HAL adsorbent calculated by Equation (17) determined with the PD ILC method.

Adsorbate	PH	2CPH	3CPH	4CPH	24DCPH	246TCPH
*T* (K)	*K* (cm^3^∙mg^−1^)^1/n^					
298	0.201	0.031	0.018	0.067	0.051	0.031
303	0.184	0.025	0.014	0.055	0.041	0.025
313	0.160	0.019	0.011	0.043	0.030	0.021

**Table 7 materials-13-03309-t007:** Verification of adsorption enthalpy for phenol, 2-chlorophenol, 3-chlorophenol, 4-chlorophenol, 2-, 4-dichlorophenol, 2-, 4-, 6-trichlorophenol on HDTMA/HAL adsorbent using Boudart’s rules.

Adsorbate	PH	2CPH	3CPH	4CPH	24DCPH	246TCPH
Adsorption Enthaply Δ*H*_a_ (kJ∙mol^−1^)	−11.6	−23.3	−26.3	−22.9	−26.1	−18.4
Standard Adsorption Entropy $\Delta S_{a}^{o}$ (J∙mol^−1^K^−1^)	−25.7	−49.3	−54.8	−53.3	−63.9	−32.8
Standard Entropy ^a^ $S_{298}^{o}$ (J∙mol^−1^K^−1^)	143.2	187.4	160.2	163.3	183.1	122.8
Boudart’s rules ^b^						
$\Delta S_{a}^{o}$ < 0	−6.2 < 0	−12.8 < 0	−13.2 < 0	−12.8 < 0	−15.5 < 0	−7.8 < 0
$\left|\Delta S_{a}^{o}\right|$ < $S_{298}^{o}$	|−6.2| < 34.3	|−12.8| < 44.8	|−13.2| < 38.3	|−12.8| < 39.1	|−15.5| < 43.8	|−7.8| < 29.4

^a^ Data source: Thermodynamic Data Centre; ^b^ Values of the adsorption enthalpy were calculated in cal/mol units while values of the standard entropy and adsorption entropy were calculated in cal/(mol K) units.

**Table 8 materials-13-03309-t008:** The number of adsorbate milligrams adsorbed per unit mass of adsorbent for phenol. 2-chlorophenol, 3-chlorophenol, 4-chlorophenol, 2-, 4-dichlorophenol and 2-, 4-, 6-trichlorophenol on the HDTMA/HAL adsorbent determined with the BC ILC method.

Adsorbate	PH	2CPH	3CPH	4CPH	24DCPH	246TCPH
*T* (K)	*a* (mg∙g^−1^)					
298	34.5	18.6	8.1	11.8	6.9	3.8
303	29.8	14.3	7.9	10.2	6.4	3.6
313	17.6	12.9	7.1	9.3	6.2	3.3

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
