# Peer review of "Adsorption of Phenol and Chlorophenols by HDTMA Modified Halloysite Nanotubes"

_materials, 2020, doi:10.3390/ma13153309_

Round 1
Reviewer 1 Report
The paper is dedicated to the adsorption of chlorophenols on the cationic surfactant-modified halloysite. The adsorption was studied by the inverse chromatography method. However, some details need to be clarified, and some mistakes corrected. The manuscript needs some copyediting.
Title:
In Title and Abstract, the HDTMA abbreviation is used without the definition.
Abstract:
The last sentence should be rephrased for clarity. Now it reads as “…structures… showed that… charge distributions… were used to explain…”, which is murky and confusing.
Introduction:
Introduction should be more focused on the adsorption of various hydrophobic substances on the surfactant-modified halloysite and other clays. Some literature suggestions:
1) Cavallaro et al., 2019 DOI:10.1039/c9ra08230a;
2) Wang et al., 2018 DOI:10.1016/j.jcis.2018.02.020;
3) Shen & Gao, 2019 DOI:10.1016/j.cej.2019.121910;
Materials and methods:
The purity of the used halloysite should be estimated. WAXS could show the impurities of other minerals, and XRF analysis could show the heavy-metal impurities that could affect the adsorption properties of the halloysite.
Results and Discussion:
Computational Calculations:
The authors suppose that the phenol compounds are adsorbed onto the double layer of HDTMA by electrostatic interactions. It would be logical to model these complexes by DFT calculations and to estimate the binding energy for the studied chlorophenols. Then these values could be compared with the adsorption constants.
SEM studies:
In Figure 4, the presented images show only a couple of nanotubes. It would be better to provide lower magnification images to show how many nanotubes are in the samples.
IGC technique:
Why are Eqs. (1)-(6) are shown in the Results and Discussion section? Are they for the first time formulated by the authors? If not, they should be moved to the Supplementary section.
The same applies to Eqs. (7)–(14).
Figure 5 repeats the earlier published Figures: e.g., Fig.2 from Słomkiewicz et al., 2015 DOI:10.1016/j.clay.2015.06.007, and Fig.2 from Czech & Słomkiewicz, 2013 DOI:10.1016/j.chroma.2013.02.045. It should be moved to the Supplementary section, as well.
In Figure 6, the Langmuir fits are very suspicious. Why are they linear? The Langmuir equation Q=Qmax(Kc/(1+Kc)) should be suitable for fitting the presented data points.
What about the higher concentration region? Will the adsorption parameters be the same?
Table 5 is mostly technical and could be moved to the Supplementary section.
In Table 7, the second row and the fourth row present the same quantity (standard adsorption entropy, ΔS0a), but their values differ. Why?
The crucial part of the Discussion is missing: the obtained results should be compared with the adsorption characteristics of phenol and chlorophenols on other similar adsorbents.
Conclusions:
Conclusions should be refined after the amendment of the Discussion.
“Application of inverse liquid chromatography methods indicates a new experimental approach to the adsorption process.” This statement is not correct because ILC is a known method for adsorption studies (see, e.g., DOI:10.1021/la0632631, DOI:10.1016/j.clay.2015.06.007, DOI:10.1016/S0021-9673(02)00379-5, and DOI:10.1021/la0261139).
Author Response
Revision 1
The paper is dedicated to the adsorption of chlorophenols on the cationic surfactant-modified halloysite. The adsorption was studied by the inverse chromatography method. However, some details need to be clarified, and some mistakes corrected. The manuscript needs some copyediting.
Question 1
Title: In Title and Abstract, the HDTMA abbreviation is used without the definition.
Answer 1
This error was corrected in manuscript.
Question 2
Abstract: The last sentence should be rephrased for clarity. Now it reads as “…structures… showed that… charge distributions… were used to explain…”, which is murky and confusing.
Answer 2
The sentence was removed.
Question 3
Introduction: Introduction should be more focused on the adsorption of various hydrophobic substances on the surfactant-modified halloysite and other clays. Some literature suggestions:
1) Cavallaro et al., 2019 DOI:10.1039/c9ra08230a;
2) Wang et al., 2018 DOI:10.1016/j.jcis.2018.02.020;
3) Shen & Gao, 2019 DOI:10.1016/j.cej.2019.121910;
Answer 3
References 1-3 have been added to the Introduction. We also added Table 1. Adsorption capacities of various adsorbents for phenol and its chloro derivatives.
Question 4
Materials and methods: The purity of the used halloysite should be estimated. WAXS could show the impurities of other minerals, and XRF analysis could show the heavy-metal impurities that could affect the adsorption properties of the halloysite.
Answer 4
We have introduced the following part: “Chemical composition of halloysite obtained by the WDXRF analysis [41] is following: Al2O3 42.3%, SiO2 50.7%, Fe2O3 2.99 %, TiO2 1.41%, CaO 0.29%, MgO 0.06%, Na2O 0.12%, K2O 0.04%, P2O5 0.51%, SO3 0.17%. In the diffractograms of the halloysite sample (XRD method) the peaks of the following minerals were identified: halloysite, kaolinite, hematite, calcite [41]” to the Chapter 3.1. Characterization of the HDTMA/halloysite adsorbent.
Question 5
Results and Discussion:
Computational Calculations:
The authors suppose that the phenol compounds are adsorbed onto the double layer of HDTMA by electrostatic interactions. It would be logical to model these complexes by DFT calculations and to estimate the binding energy for the studied chlorophenols. Then these values could be compared with the adsorption constants.
Answer 5
Czapter 5.1 Computational calculations’ was removed from manuscript according to reviewer 4 comments.
Question 6
SEM studies: In Figure 4, the presented images show only a couple of nanotubes. It would be better to provide lower magnification images to show how many nanotubes are in the samples.
Answer 6
In the Chapter 3.1. Characterization of the HDTMA/halloysite adsorbent we changed the Figure 5 – we added TEM image of halloysite sample for better clarity and new sentence in the manuscript: “SEM and TEM images (Fig. 5) confirm the tubular structure of halloysite (HAL) sample. Modification of halloysite with HDTMA does not change the mineral structure significantly.”
Question 7
IGC technique: Why are Eqs. (1)-(6) are shown in the Results and Discussion section? Are they for the first time formulated by the authors? If not, they should be moved to the Supplementary section.
Answer 7
The manuscript was changed. All information about ILC methods have been placed in the Chapter 2.4. Inverse Liquid Chromatography Methods - Adsorption measurements by the PD ILC Method and -Adsorption measurements by the BC ILC Method. Equation 1-2 in this Chapter were published in Ref. [45]. Equations 3-8 are presented in this Chapter for the first time.
Question 8
The same applies to Eqs. (7)–(14).
Answer 8
The answer, as above.
Question 9
Figure 5 repeats the earlier published Figures: e.g., Fig.2 from Słomkiewicz et al., 2015 DOI:10.1016/j.clay.2015.06.007, and Fig.2 from Czech & Słomkiewicz, 2013 DOI:10.1016/j.chroma.2013.02.045. It should be moved to the Supplementary section, as well.
Answer 9
Figure 1 (Figure 5 in old version) differ from Figures 2 presented in the articles: Czech & Słomkiewicz, 2013 DOI:10.1016/j.chroma.2013.02.045 and in Słomkiewicz et al., 2015 DOI:10.1016/j.clay.2015.06.007). In our manuscript Figure 1 illustrates the method of the additional measurement of non-adsorbed substance in the reference column for eliminating the diffusion influence on the peak shape. This method was not be published yet.
Question 10
In Figure 6, the Langmuir fits are very suspicious. Why are they linear? The Langmuir equation Q=Qmax(Kc/(1+Kc)) should be suitable for fitting the presented data points.
Answer 10
Fit of adsorption equations to the experimental data was completed with the Levenberg–Marquardt least-squares method using the Origin Microcal. The linear fit of equation 16 to the experimental data was obtained from Origin Microcal. It is probably caused the low concentrations of adsorbates.
Question 11
What about the higher concentration region? Will the adsorption parameters be the same?
Answer 11
The peaks obtained by the ILC method for adsorption at low concentration have the asymmetric shape. The increase of the adsorbate concentration leads to overload of the column and then the peak can change its shape. In this situation the adsorption parameters may change.
Question 12
Table 5 is mostly technical and could be moved to the Supplementary section.
Answer 12
Table 5 shows results of calculations by the least-squares method adjusted to the adsorption data equations of adsorption. As can be seen from the Fig. 5, only the curves determined by Equation (17-19) coincided with the experimental points. The remaining curve, determined by equation 16, did not overlap with the experimental points. The data presented in Table 5 allow to determine the best fit equation for 17.
We propose to leave it in the Chapter 3.1. Characterization of the HDTMA/halloysite adsorbent.
Question 13
In Table 7, the second row and the fourth row present the same quantity (standard adsorption entropy, ΔS0a), but their values differ. Why?
Answer 13
The standard adsorption entropy was in units [J∙mol-1K-1]. The Boudart’s rules was published (M. Boudart, D. E. Mears, M. A, Vannice, Kinetics of heterogeneous catalytic reactions Ind. Chim. Belg. 1967, 32, 281-284) many years ago and standard adsorption entropy was in units [cal/(mol K)]. Therefore we leave it and explanation was in the bottom of the Table 7.
Question 14
The crucial part of the Discussion is missing: the obtained results should be compared with the adsorption characteristics of phenol and chlorophenols on other similar adsorbents.
Answer 14
The adsorption of phenol and chlorophenols on other similar adsorbents are presented in the Introduction (Table 1). These results were obtained in batch system. We have presented the adsorption results in flow chromatographic system. It is difficult to compare the results obtained with these methods.
Question 15
Conclusions: Conclusions should be refined after the amendment of the Discussion.
Answer 15
Conclusions were changed.
Question 16
“Application of inverse liquid chromatography methods indicates a new experimental approach to the adsorption process.” This statement is not correct because ILC is a known method for adsorption studies (see, e.g., DOI:10.1021/la0632631, DOI:10.1016/j.clay.2015.06.007, DOI:10.1016/S0021-9673(02)00379-5, and DOI:10.1021/la0261139).
Answer 16
The sentence “Application of inverse liquid chromatography methods indicates a new experimental approach to the adsorption process” was repaced by the sentence: “Application of inverse liquid chromatography methods (Peak Division and the Breakthrough Curves) indicates a new experimental approach to the adsorption process”.

Reviewer 2 Report
Please see the attached file

Author Response
Revision 4
Question 1
In this paper the adsorption of phenol, 2-, 3-, 4-chlorophenol, 2,4-dichlorophenol, and 2,4,6-trichloro-11 phenol on HDTMA/halloysite nanocomposite was studied by means of several analytical techniques. Besides, the authors tried to give an interpretation of the observed adsorption behaviour with an appreciable scientific judgement. However, the theoretical/computational paragraph introduced does not appear to be up to the overall level of the work. The electron density/potential analysis is in fact something of rigorous and cannot be solved by saying “General electron density distribution did not change, contrary to the values of the partial charges…” (page 3, line 136. Moreover, at page 4,line 143 the sentence “….which allowed to conclude that in this case adsorption was strongly associated with the electron distribution” seems to be in contrast to the previous assertion…).
The rearrangement of the electron density (and, consequently, of any potential associated) in a molecule/crystal can be rigorously described by means of theoretical tools such as the “Bader’s topological analysis”, which allows to get a lot of information such as the recognition of reactivity sites in a molecule, the evaluation of the effective charge (and a number of other quantities) in an atom basin… and so on.
Moreover, if the object of the computational study is whole system halloysite+phenol (which would be pleasing), Bader’s analysis allows to verify (in a rigorous framework) if two atoms are bonded or not.
In the light of these considerations, I think that the manuscript could improve much if either ) the ‘5.1 Computational calculations’ is removed or ii) the ‘5.1 Computational calculations’ is renamed into ‘Computational approach’, or ‘Ab-initio simulations’ or something similar and rewritten by adding a rigorous analysis of the electron density (and its derivatives) and/or the electrostatic potential.
Answer 1
Part 5.1 Computational calculations’ was removed from manuscript.

Reviewer 3 Report
Please see the attached.

Author Response
Revision 2
The submitted work reports the adsorption efficiency of phenol, 2-, 3-, 4-chlorophenol, 2,4-dichlorophenol, and 2,4,6-trichloro phenol on HDTMA/halloysite nanocomposite. Specific surface energy heterogeneity profiles and acid base properties of halloysite and HDTMA/halloysite nanocomposite are shown herein. The obtained experimental adsorption data were well represented by the multi-center adsorption model. The DFT calculation showed charge distributions of the adsorbate molecules and the related results were used to explain the adsorption mechanism of phenol and chlorophenols on the HDTMA/halloysite nanocomposite. Overall, the manuscript is well constructed. The discussions based on the rigorous calculation are fine. This work is of potential interest for the nanomaterials community. The manuscript can be accepted for publication after minor revisions.
Question 1
Although the introduction section highlighted the importance of the proposed absorbents for phenol and chlorophenols removal applications, the systematic comparisons of the phenol and/or chlorophenols removal efficiency of the as-synthesized nanocomposite herein with other reported material systems are not summarized in this study. To enhance the impact of this study, a systematic comparison of the current results with other similar reported works (preferred to use a comparison Table) should be highlighted in the revised version.
Answer 1
We added the Table 1. Adsorption capacities of various adsorbents for phenol and its chloro derivatives to the Introduction in the manuscript.
Question 2
The scale bar of the SEM images should be remade for clarity. Moreover, the average particle size should be provided to support their BET results.
Answer 2
In the Chapter 3.1. Characterization of the HDTMA/halloysite adsorbent we changed the Figure 5 – we added TEM image of halloysite sample for better clarity and new sentence in the manuscript: “SEM and TEM images (Fig. 5) confirm the tubular structure of halloysite (HAL) sample. Modification of halloysite with HDTMA does not change the mineral structure significantly.”
Question 3
Discussions on morphology and structural changes of the HDTMA/halloysite nanocomposite are limited herein, pleased enhance this section.
Answer 3
We have introduced the following part: “Chemical composition of halloysite obtained by the WDXRF analysis [41] is following: Al2O3 42.3%, SiO2 50.7%, Fe2O3 2.99 %, TiO2 1.41%, CaO 0.29%, MgO 0.06%, Na2O 0.12%, K2O 0.04%, P2O5 0.51%, SO3 0.17%. In the diffractograms of the halloysite sample (XRD method) the peaks of the following minerals were identified: halloysite, kaolinite, hematite, calcite [41]” to the Chapter 3.1. Characterization of the HDTMA/halloysite adsorbent.

Reviewer 4 Report
This manuscript describes a direct method of measurement of adsorption of some halophenols on halloysite modified with the surfactant HDTMA by chromatography. Initially the topics seemed to have novelty and interesting, however after reading the manuscript a sad frustration is obtaining. No novelty is reported and a mixture of unconnected results are described. The main part of manuscript is focussed on technical developments of chromatographic analysis. Therefore, I consider that this manuscript cannot be published in the Materials journal and I recommend to submit it to a journal more oriented to analytical chemistry or Chromatography after addressing the following comments:
General: some spelling and grammar mistakes. A general revision would be welcome.
Lines 33: rewrite this sentence.
Line 35: reedit the style. This sentence is difficult to understand.
Line 39: this sentence is obvious.
Line 46: define HDTMA the first time where it appears.
Section 2.2: describe the mineralogy of the natural sample of halloysite, or explain the reference where is described, percentage of other minerals, kaolinite, smectites, carbonates, quartz, etc…
Section 5.1: the technical aspects should be put before the Results section. This section does not give any new knowledge and it should be deleted completely. The technical aspects are not described properly. The definition of partial charges is not included, which kind of net atomic charges are used. No dispersion correction is used, whereas the long range interactions are important for the interactions of this kind of molecules. It looks like that the authors have used this section without understanding these calculations. Besides, they do not use these theoretical results for the rest of results.
Lines 164-167 and table 2: Some mistakes in the data in Table and text are found. Please rewrite this.
Nothing new is presented in the FT-IR and SEM results and both should be deleted. Authors compare HAL with HDTMA/HAL but not with the adsorbates and later they do not compare both solids in the chromatography analysis.
Lines 183-232.
Most of this should be before the results, since they are technical aspects of the study. Please explain strange concept s as ‘inclination coefficient’. Authors cite NMR data but do not explain it.
Section 6: This is more for a chromatography journal. No equilibrium is reached to evaluate the actual adsorption process. No comparison between HAL and HDTMA/HAL is presented.
Finally, I cannot recommend it for publishing in Materials journal.
Author Response
Revision 3
This manuscript describes a direct method of measurement of adsorption of some halophenols on halloysite modified with the surfactant HDTMA by chromatography. Initially the topics seemed to have novelty and interesting, however after reading the manuscript a sad frustration is obtaining. No novelty is reported and a mixture of unconnected results are described. The main part of manuscript is focussed on technical developments of chromatographic analysis. Therefore, I consider that this manuscript cannot be published in the Materials journal and I recommend to submit it to a journal more oriented to analytical chemistry or Chromatography after addressing the following comments:
Question 1
General: some spelling and grammar mistakes. A general revision would be welcome.
Lines 33: rewrite this sentence.
Line 35: reedit the style. This sentence is difficult to understand.
Line 39: this sentence is obvious.
Line 46: define HDTMA the first time where it appears.
Answer 1
Line 33: the sentence: “The number and position of chlorine atoms in the benzene ring relative to the hydroxyl group influences the toxicity of chlorophenols [3]”
has been rewritten as “The toxicity of chlorophenols is affected by the number and position of chlorine atoms in the benzene ring [3]”.
Line 35: the sentence: “Their environmental sustainability, poor biodegradability, unpleasant odor, and they impart taste to drinking water cause the need to remove these compounds from the environment” has been shortened: “Their environmental sustainability, poor biodegradability, unpleasant odor cause the need to remove these compounds from the environment”.
Line 46: HDTMA was defined.
Question 2
Section 2.2: describe the mineralogy of the natural sample of halloysite, or explain the reference where is described, percentage of other minerals, kaolinite, smectites, carbonates, quartz, etc…
Answer 2
We have introduced the following part: “Chemical composition of halloysite obtained by the WDXRF analysis [41] is following: Al2O3 42.3%, SiO2 50.7%, Fe2O3 2.99 %, TiO2 1.41%, CaO 0.29%, MgO 0.06%, Na2O 0.12%, K2O 0.04%, P2O5 0.51%, SO3 0.17%. In the diffractograms of the halloysite sample (XRD method) the peaks of the following minerals were identified: halloysite, kaolinite, hematite, calcite [41]” to the Chapter 3.1. Characterization of the HDTMA/halloysite adsorbent.
Question 3
Section 5.1: the technical aspects should be put before the Results section. This section does not give any new knowledge and it should be deleted completely. The technical aspects are not described properly. The definition of partial charges is not included, which kind of net atomic charges are used. No dispersion correction is used, whereas the long range interactions are important for the interactions of this kind of molecules. It looks like that the authors have used this section without understanding these calculations. Besides, they do not use these theoretical results for the rest of results.
Answer 3
Part 5.1 Computational calculations’ was removed from manuscript according to reviewer 4 comments.
Question 4
Lines 164-167 and table 2: Some mistakes in the data in Table and text are found. Please rewrite this.
Answer 4
Data in the Table 2 were corrected.
Question 5
Nothing new is presented in the FT-IR and SEM results and both should be deleted. Authors compare HAL with HDTMA/HAL but not with the adsorbates and later they do not compare both solids in the chromatography analysis.
Answer 5
ATR FT-IR spectra confirmed the presence of HDTMA on the halloysite surface, SEM and TEM images confirmed the tubular structure of halloysite.
Question 6
Lines 183-232.
Most of this should be before the results, since they are technical aspects of the study. Please explain strange concept s as ‘inclination coefficient’. Authors cite NMR data but do not explain it.
Answer 6
The ‘inclination coefficient’ has been replaced by “slope coefficient”. NMR data was removed.
Question 7
Section 6: This is more for a chromatography journal. No equilibrium is reached to evaluate the actual adsorption process. No comparison between HAL and HDTMA/HAL is presented.
Answer 7
Adsorption measurements carried out for phenol and chlorophenols on halloysite and halloysite/HDTMA adsorbents showed that unmodified halloysite does not practically adsorb these compounds. Therefore, further studies were carried out for the halloysite/HDTMA adsorbent.
The adsorption of phenol and chlorophenols on other similar adsorbents presented in the Introduction (Table 1) was performed in batch system. We have presented the adsorption results in flow chromatographic system. It is difficult to compare the results obtained with these methods.

Round 2
Reviewer 1 Report
The authors improved the paper, but several serious issues remain.
Question 10: In Figure 6, the Langmuir fits are very suspicious. Why are they linear? The Langmuir equation Q=Qmax(Kc/(1+Kc)) should be suitable for fitting the presented data points.
Answer 10: Fit of adsorption equations to the experimental data was completed with the Levenberg–Marquardt least-squares method using the Origin Microcal. The linear fit of equation 16 to the experimental data was obtained from Origin Microcal. It is probably caused the low concentrations of adsorbates.
Comment: There are no units indicated for ai (the quantity shown in y-axes in Fig.6). If we assume that this is the same ai as in Eq.1, its units should be (mg adsorbate/mg adsorbent). But then all the fits by Eqs.16–19 implicitly assume that the maximum adsorption is 1. This explains why the authors could not fit the Langmuir model properly. In Eqs.16–19, there should be amax parameter included, as in Langmuir model: ai = amax×K×ci/(1+ K×ci). Then the Langmuir fits would be properly adjusted.
One small example: Langmuir fits for Fig.6 in the revised manuscript by three different linearization approaches and non-linear fitting (see the picture attached). Note that all four fits are not linear.
The authors should include the dataset for adsorption studies in the Supplementary section; then, the readers and reviewers could check the model fits.
Question 11: What about the higher concentration region? Will the adsorption parameters be the same?
Answer 11: The peaks obtained by the ILC method for adsorption at low concentration have the asymmetric shape. The increase of the adsorbate concentration leads to overload of the column and then the peak can change its shape. In this situation the adsorption parameters may change.
Comment: If the proposed method cannot provide the information for the higher concentration range, then it should be complemented with other adsorption studies.
Question 14: The crucial part of the Discussion is missing: the obtained results should be compared with the adsorption characteristics of phenol and chlorophenols on other similar adsorbents.
Answer 14: The adsorption of phenol and chlorophenols on other similar adsorbents are presented in the Introduction (Table 1). These results were obtained in batch system. We have presented the adsorption results in flow chromatographic system. It is difficult to compare the results obtained with these methods.
Comment: If the presented results could not be compared with previously published results, then the proposed method is of little value. The adsorption studies should be complemented with other methods.

Author Response
Question 10: In Figure 6, the Langmuir fits are very suspicious. Why are they linear? The Langmuir equation Q=Qmax(Kc/(1+Kc)) should be suitable for fitting the presented data points.
Answer 10: Fit of adsorption equations to the experimental data was completed with the Levenberg–Marquardt least-squares method using the Origin Microcal. The linear fit of equation 16 to the experimental data was obtained from Origin Microcal. It is probably caused the low concentrations of adsorbates.
Comment: There are no units indicated for ai (the quantity shown in y-axes in Fig.6). If we assume that this is the same ai as in Eq.1, its units should be (mg adsorbate/mg adsorbent). But then all the fits by Eqs.16–19 implicitly assume that the maximum adsorption is 1. This explains why the authors could not fit the Langmuir model properly. In Eqs.16–19, there should be amax parameter included, as in Langmuir model: ai = amax×K×ci/(1+ K×ci). Then the Langmuir fits would be properly adjusted.
One small example: Langmuir fits for Fig.6 in the revised manuscript by three different linearization approaches and non-linear fitting (see the picture attached). Note that all four fits are not linear.
The authors should include the dataset for adsorption studies in the Supplementary section; then, the readers and reviewers could check the model fits.
Answer to Comment
In PD method, adsorbate samples are introduced into the liquid phase stream flowing through the column. As a result of repeated adsorption and desorption of the adsorbate, a temporary equilibrium is established.
In the infinite dilution method is not possible to achieve full adsorption equilibrium or it is difficult to determine when such equilibrium is reached. Based on the Equation (1), the quantity of the adsorbed substance i on adsorbent a can be determined at the equilibrium concentration of the adsorbate in the mobile phase, i.e. the adsorption isotherm, (Paryjczak, T. Gas Chromatography in Adsorption and Catalysis; Ellis Harwood Limited Publisher: Chichester, 1986).
The form of adsorption equation and equilibrium constant value can be determined from the function ai = f(ci), but knowing what the measurement conditions are, it is not possible to determine the adsorption capacity of adsorbent. It is well known that a variety of Langmuir-type equation forms can be used at a low adsorbate concentration, which was used in the presented measurement method. These equations (Eqs.16–18) allow for a more precise determination of the adsorption model: one-center adsorption without dissociation, two-center adsorption without dissociation, n-center adsorption without dissociation, and two-center adsorption with dissociation (Prokop, Z., & Setinek, K. (1974) Collect. Czech. Chem. Commun. 39, 1253–1263, Thanh, L.N., Setinek, K., & Beranek, L. (1972) Collect. Czech.Chem. Commun. 37, 3878–3884).
The equations (Eqs.16–18) have a simplified form, because the adsorption capacity cannot be determined on their basis. We successfully used this method in article: B. Szczepanik, P. M. Słomkiewicz, M. Garnuszek, Determination of adsorption isotherms of aniline and 4-chloroaniline on halloysite adsorbent by inverse liquid chromatography, Applied Clay Science 114 (2015) 221–228 and P. M. Słomkiewicz, B. Szczepanik, M. Garnuszek, P.Rogala, Z. Witkiewicz, Determination of Adsorption Equations for Chloro Derivatives of Aniline on Halloysite Adsorbents Using Inverse Liquid Chromatography, Journal of AOAC International Vol. 100, No. 6, 2017.
The dataset for adsorption studies in the Supplementary section was attached.
Question 11: What about the higher concentration region? Will the adsorption parameters be the same?
Answer 11: The peaks obtained by the ILC method for adsorption at low concentration have the asymmetric shape. The increase of the adsorbate concentration leads to overload of the column and then the peak can change its shape. In this situation the adsorption parameters may change.
Comment: If the proposed method cannot provide the information for the higher concentration range, then it should be complemented with other adsorption studies.
Answer to Comment
In this manuscript we have presented two complementary measurement methods. In PD method for low adsorbate concentrations the adsorption model and corresponding adsorption equation can be determined. This method can be included in the inverse liquid chromatography methods at infinite dilution. In BC method for high adsorbate concentrations the adsorption volume can be determined. This method can be included in the inverse liquid chromatography methods at finite concentration. Both measurements using these methods can be performed on the same liquid chromatograph and they enable measurements in the lower and higher concetration region.
Question 14: The crucial part of the Discussion is missing: the obtained results should be compared with the adsorption characteristics of phenol and chlorophenols on other similar adsorbents.
Answer 14: The adsorption of phenol and chlorophenols on other similar adsorbents are presented in the Introduction (Table 1). These results were obtained in batch system. We have presented the adsorption results in flow chromatographic system. It is difficult to compare the results obtained with these methods.
Comment: If the presented results could not be compared with previously published results, then the proposed method is of little value. The adsorption studies should be complemented with other methods.
Answer to Comment
We added the text in Conclusions:
Phenol and chlorophenols adsorption measurements were carried out in the flow system, mainly on carbon adsorbents and resins [61-64]. In these works the breakthrough curve method was used to determine the adsorption capacity of adsorbent.
According to our knowledge we applied the Peak Division and the Breakthrough Curves ILC methods for the adsorption phenol and chlorophenols on halloysite modified by HDTMA. The first method can be used to determine the form of the adsorption equation and the second one can be used to determine the adsorption capacity of adsorbent. In addition, these methods provides experimental simplicity combined with low adsorbate and solvent consumption.

Reviewer 2 Report
now it is ok for me
Author Response
Thank you for your careful reviews
